# Activation of the ROS/CncC Signaling Pathway Regulates Cytochrome P450 *CYP4BQ1* Responsible for (+)-α-Pinene Tolerance in *Dendroctonus armandi*

**DOI:** 10.3390/ijms231911578

**Published:** 2022-09-30

**Authors:** Bin Liu, Ming Tang, Hui Chen

**Affiliations:** 1College of Forestry, Northwest A&F University, Xianyang 712100, China; 2State Key Laboratory for Conservation and Utilization of Subtropical Agro-Bioresources, Guangdong Laboratory for Lingnan Modern Agriculture, College of Forestry and Landscape Architecture, South China Agricultural University, Guangzhou 510642, China

**Keywords:** *Dendroctonus armandi*, phytochemical tolerance, cytochrome P450, oxidative burst, CncC pathway

## Abstract

Bark beetles mainly rely on detoxification enzymes to resist the host tree’s defense against oleoresin terpenes. Cytochrome P450 enzymes (CYPs) play an important role in the detoxification of plant allelochemicals and pesticides in insect. One P450 gene (*DaCYP4BQ1*) is associated with the response of (+)-α-pinene in *Dendroctonus armandi*. However, the regulatory mechanism of this P450 gene response to (+)-α-pinene is still unknown. In this study, spatiotemporal expression profiling indicated that *CYP4BQ1* was highly expressed in adult and larval stages of *D. armandi*, and it was predominantly expressed in fat body, midgut, and Malpighian tubules of adults. Moreover, the expression of *CYP4BQ1* significantly increased after exposure to (+)-α-pinene, and depletion of it decreased the tolerance of adults to (+)-α-pinene. In addition, (+)-α-pinene treatment induced the expression of the transcription factors cap ‘n’ collar isoform C (*CncC*) and its binding factor muscle aponeurosis fibromatosis (*Maf*), elevated the level of hydrogen peroxide (H_2_O_2_), and increased the activities of antioxidant enzymes. Silencing *CncC* suppressed *CYP4BQ1* expression and enhanced the susceptibility of beetles to (+)-α-pinene. Similarly, application of the reactive oxygen species (ROS) scavenger N-acetylcysteine reduced the production and accumulation of H_2_O_2_, suppressed the expression of *CncC, Maf*, and *CYP4BQ1* and led to decreased tolerance of adults to (+)-α-pinene. In contrast, ingestion of the CncC agonist curcumin elevated *CYP4BQ1* expression and enhanced (+)-α-pinene tolerance. The results demonstrate that, in *D. armandi*, (+)-α-pinene induces *CYP4BQ1* via activation of the ROS/CncC signaling pathway.

## 1. Introduction

The Chinese white pine beetle, *Dendroctonus armandi,* is a destructive pest of coniferous forests in the middle Qinling and Bashan Mountains of China. It not only attacks healthy Chinese white pine trees, *Pinus armandii*, that are more than 30 years old, but also attracts other pests to the host trees. This has led to the destruction of the forest ecological system and caused heavy economic losses [1]. Bark beetles complete their life cycle under the bark of host trees unless they spread to new trees and reproduce. One important stage of bark beetles in the life cycle is the host colonization period, during which they need overcome the defense of pine tree to reproduce successfully [2].

The resistance of *Pinus armandii* to bark beetles mainly depends on the composition and induced chemical and physical defense, and the composition and induced oleoresin terpenes are the main defense components [3]. Oleoresin is a complex composed of dozens of monoterpenes, diterpenes, and a few sesquiterpenes [4]. α-Pinene is the primary component of oleoresin terpenes in *P*. *armandii* [3], and (+)-α-pinene showed high fumigant toxicity to the adult stages of *D. armandi* [5]. In addition, it has certain toxicity to many kinds of organisms [6,7,8,9,10]. However, in the long-term competition between herbivorous insects and plants, herbivorous insects have evolved a variety of detoxification enzymes, which can decompose various plants secondary metabolites/insecticides, making insects have strong adaptability to these chemical components [11]. Among these detoxification enzymes, cytochrome P450 monooxygenases (CYPs) play a vital role in the catabolism of heterologous substances.

CYPs have long been recognized for their ability to mediate the complex insertion of molecular oxygen into different substrates. A large number of studies have indicated that insect P450-mediated mechanisms are most often used to confer insect resistance to pesticides/phytochemicals [12]. For example, transcriptome analysis of *Spodoptera litura* revealed that the extended P450 genes could be induced after imidacloprid exposure, and the silencing of the corresponding P450 gene enhanced the sensitivity to imidacloprid [13]. In addition, the expression of *CYP4Q3* was significantly induced by imidacloprid, and depletion of it increased the susceptibility to insecticide in *Leptinotarsa decemlineata* [14]. Similarly, gossypol upregulated the expression of *CYP6DA2*, and knockdown of its expression significantly enhanced the toxicity of gossypol to *Aphis gossypii* [15]. Moreover, in *Dendroctonus rhizophagus,* the genes in the CYP4 family play a key role in the detoxification of monoterpenes produced by the host [16]. In *Tribolium castaneum,* the expressions of three *CYP4* genes were significantly induced after exposure to eugenol, and their silencing decreased the tolerance of beetles to eugenol [17]. Although many studies have shown that pesticides/phytochemicals upregulate insect detoxification genes, little is known about the regulatory mechanism of these genes, especially the first step of the signal cascade.

Understanding the regulatory network that coordinates the expression of detoxification gene is a necessary condition for breaking down the resistance [18]. As for the regulatory mechanism of P450 overexpression in resistant strains and/or induction after exposure to pesticides/phytochemicals, transcription factors (TFs) and cis-acting elements have been proven to be the most common [19,20,21]. TFs sensing xenobiotic stress are called as xenobiotic sensors, which include three kinds of TF superfamilies, such as basic-helix-loop-helix/Per-ARNTSim (bHLH-PAS), basic leucine zipper (bZIP) proteins, and nuclear receptors (NRs) [22]. Among these, the cap ‘n’ collar isoform C (CncC) of bZIP is one of the important regulators of insect response to exogenous substances, which is homologous with nuclear factor erythroid 2-related factor (Nrf2) in mammals [19,23]. Under normal conditions, CncC is bound to Kelch-like ECH-associated protein 1 (Keap1, the ubiquitin ligase) and retained in the cytoplasm anchored to actin filaments. However, upon xenobiotic stress, the generated reactive oxygen species (ROS) induce the separation of CncC from the tight CncC/Keap1 complex, making CncC translocate to the nucleus and heterodimerize with its partner muscle aponeurosis fibromatosis (Maf). Subsequently, the CncC/Maf heterodimer binds with antioxidant response elements (AREs), which are located in the promoter region of detoxification genes, finally initiating the expression of P450 genes [24,25]. So far, the information on the detailed mechanisms of the ROS/CncC signaling pathway-mediated regulation of insect detoxification gene transcription has mainly come from studies on insecticide resistance/tolerance [19]. In addition, the indirect indications of the ROS/CncC signaling pathway in phytochemical inducibility of detoxification genes were also obtained from *L. decemlineata*, in which the potato leaf extract induced four P450 genes. In addition, gossypol upregulated the expression of *CYP6DA2* in *A. gossypii* [15,26]. These studies indicated that RNA interference (RNAi)-mediated silencing of *CncC* reduced P450 gene expression. However, more work is needed to demonstrate whether these phytochemicals act as the ROS elicitors to induce the nuclear translocation of CncC and if phytochemical induction of these P450s is CncC-mediated.

As pesticides have never been used against bark beetles in our sampling site, the host’s own chemical defense is the main pressure on *D. armandi*. Sequence amplifications, transcriptional enhancements, and coding mutations in genes encoding CYPs, glutathione *S*-transferases (GSTs), and esterases (ESTs) are resistance mechanisms in insects [27]. CYP is the main research focus for understanding the molecular mechanism of host selection and colonization behavior of the bark beetle [28]. In the previous study of our laboratory, we found that four genes in the CYP4 subfamily from *D. armandi* were highly expressed after the treatment of host defensive chemicals. However, whether they are also involved in tolerance of (+)-α-pinene, which is the primary component of oleoresin terpenes in *P*. *armandii,* has not been reported. Here, in the present study, we found that one of the four genes, *CYP4BQ1*, was significantly expressed after (+)-α-pinene exposure. However, the relevant regulatory pathways and detailed mechanisms involved in the induction of *D. armandi CYP4BQ1* following exposure to (+)-α-pinene are not clear. Therefore, we functionally confirm the causal role of *CYP4BQ1* in resistance to (+)-α-pinene and investigate the factors driving its upregulation after exposure to (+)-α-pinene. Altogether, this study can help to understand the response mechanisms of *D. armandi* to (+)-α-pinene and provide evidence for pest control.

## 2. Results

### 2.1. (+)-α-Pinene Exposure Induces DaCYP4BQ1 Expression

The expression of four CYP4 subfamily P450 genes was compared with qRT-PCR analysis between (+)-α-pinene-treated (LC_50_) and control insects. Among these four P450s, only *CYP4BQ1* expression of *D. armandi* adults was significantly induced at 72 h post exposure to (+)-α-pinene (Figure 1).

### 2.2. Spatiotemporal Expression Pattern of CYP4BQ1

*CYP4BQ1* was expressed in all developmental stages of *D. armandi*. It was expressed the highest in adult stage, followed by larval stage, and the lowest in early pupae (Appendix A). Moreover, as for the adult stage, the expression level of *CYP4BQ1* in females was higher than that in males in the attacking adults (Appendix A). *CYP4BQ1* was expressed at different levels and with occasional sex differences among the different tissues. Specifically, *CYP4BQ1* was predominantly expressed in the fat body, midgut, and Malpighian tubules, while little expression was found in other parts (Appendix A). Moreover, the expression level of *CYP4BQ1* in females was higher than that in males in the midgut and fat body (Appendix A).

### 2.3. CYP4BQ1 Participates in Adults’ Tolerance to (+)-α-Pinene

We further analyzed the transcriptional responses of *CYP4BQ1* to (+)-α-pinene at different timepoints. (+)-α-Pinene exposure induced the expression of *CYP4BQ1* by 3.04- and 6.82-fold at 48 and 72 h in male adults, respectively, when compared with controls (Figure 2A). The equivalent in female adults was 5.11- and 8.16-fold, respectively (Figure 2E). To further determine the function of *DaCYP4BQ1* on beetle resistance to (+)-α-pinene, RNAi of this P450 gene was performed. The expression level of *CYP4BQ1* in adults was significantly downregulated at 24, 48, and 72 h after dsRNA injection when compared with the control group, except for male adults at 24 h (Figure 2B**)**. In addition, the expression level of *CYP4BQ1* was reduced most in males and females, reaching 51.2% and 67.7%, respectively (Figure 2B,F). However, silencing *CYP4BQ1* did not significantly affect the expression of nontarget genes (Appendix A). This showed that the RNAi targeted to *CYP4BQ1* did not have off-target effects. The P450 enzyme activity of adults significantly decreased at 72 h after injection of dsRNA when compared with the control group, which reduced P450 activity in male and female adults by 44.7% and 56.6%, respectively (Figure 2C,G). The mortality of adults with the dsRNA injection after (+)-α-pinene exposure was significantly higher than the control, which was enhanced by 34.3% and 42.5% in male and female adults, respectively (Figure 2D,H).

### 2.4. (+)-α-Pinene Exposure Induces Expression of CncC and Maf

To further study the relationship between (+)-α-pinene and the CncC signaling pathway, the expression level of *DaCncC* was measured after (+)-α-pinene treatment. After (+)-α-pinene exposure, the expression levels of *DaCncC* and *Maf* were significantly upregulated at 48 h and 72 h in male and female adults, while the expression level of *Keap1* was not affected (Figure 3).

### 2.5. Silencing CncC Suppresses CYP4BQ1 Expression and Reduces Adults’ Tolerance to (+)-α-Pinene

The expression levels of *CncC* and *CYP4BQ1* in adults were significantly downregulated at 48 and 72 h after ds*CncC* injection when compared with the control group (Figure 4). Moreover, the expression level of *CncC* decreased most in male and female adults, reaching 53.7% and 79.2%, respectively (Figure 4A,E). That the expression level of *CYP4BQ1* was 52.7% and 55.3%, respectively (Figure 4B,F). In addition, the P450 activities of the adults’ midgut were suppressed after silencing *CncC* (Figure 4C,G). We further detected the susceptibility of adult to (+)-α-pinene after the application of RNAi. *CncC* knockdown increased mortality of male adults by 20.6 % after (+)-α-pinene exposure compared with the control group (Figure 4D), that the equivalent in female adults was 30.2% (Figure 4H).

### 2.6. (+)-α-Pinene Exposure Enhances H_2_O_2_ Generation and the Activities of Antioxidant Enzymes 

(+)-α-Pinene treatment significantly increased H_2_O_2_ generation and antioxidant enzymatic activities in the adult midgut (Figure 5). Compared with the control group, H_2_O_2_ content and antioxidant enzymatic activities of adults significantly increased at 48 and 72 h after (+)-α-pinene exposure, except for H_2_O_2_ content and POD activities in male adults at 24 h (Figure 5). In addition, H_2_O_2_ content and the activities SOD, CAT, and POD were enhanced most in male adults after (+)-α-pinene exposure, reaching 18.3%, 21.6%, 19.9% and 26.3%, respectively. The equivalent in female adults was 25.0%, 24.3%, 38.5%, and 30.4%, respectively (Figure 5).

### 2.7. Scavenging of ROS Downregulates CncC and CYP4BQ1 Expressions and Reduces Adults’ Tolerance to (+)-α-Pinene

Application of the ROS scavenger N-acetylcysteine (NAC) significantly suppressed the (+)-α-pinene-induced *CncC*, *Maf*, and *CYP4BQ1* expression in adults (Figure 6). In addition, NAC ingestion effectively inhibited the (+)-α-pinene-induced H_2_O_2_ generation (Figure 7A,D) and suppressed P450 activities (Figure 7B,E). Adults were treated with NAC after (+)-α-pinene exposure, and mortality was determined after 24 h. The susceptibility of the adults to (+)-α-pinene increased by 28.4% and 33.3% in male and female, respectively, after effective scavenging of ROS by NAC (Figure 7C,F).

### 2.8. The CncC Agonist Curcumin Induces the Expression of CYP4BQ1 and Enhances Adults’ Tolerance to (+)-α-Pinene

The mRNA levels of *CncC*, *Maf*, and *CYP4BQ1* in the CncC agonist curcumin-treated group significantly increased compared to the control (Figure 8A,B). Midgut P450 activities were significantly higher in adults treated with curcumin (Figure 8C). The mortality of male and female adults was significantly decreased by 31.6% and 18.9% (Figure 8D), respectively, after curcumin activated the CncC pathway.

## 3. Discussion

Several previous studies have indicated that CYPs are the main metabolic detoxification enzymes, which participate in the detoxification of pesticides and the development of resistance of insect populations [29,30,31]. Insect CYP genes belong to many CYP subfamilies, and their expression patterns in different tissues and developmental stages are significantly different, which provides important clues for their physiological functions [32,33]. 

In this study, survey of different stages indicated that the transcript of *CYP4BQ1* was expressed at all developmental stages of *D. armandi*. Among these, *CYP4BQ1* was expressed the highest in the adult stage, followed by the larval stage. In insects, CYP4 family members are thought to participate in the biosynthesis of endogenous complexes [34,35] and the metabolism of pheromones [36,37], in addition to insecticide/phytochemicals resistance, which suggests that *CYP4BQ1* may participate in some physiological processes regulating the development and growth of beetles. In addition, the expression of *CYP4BQ1* was further investigated in various tissues of adults, suggesting that it was expressed predominantly in the midgut and fat body. As a dynamic tissue, the fat body participates in various metabolic functions, playing a key role in the insect metabolism [38]. The midgut not only is the place where insects digest and absorb, but also has the function of resisting exogenous substances [31]. It is obvious that *DaCYP4BQ1* is involved in insect metabolism. Similar results were also found in *Manduca sexta*, where the expression levels of *CYP4L* and *CYP4M* genes were observed in the midgut and fat body, revealing that CYP4 genes highly expressed in the fat body and midgut may participate in the tissue response to exogenous substances [39].

The biological activities of α-pinene against herbivores in plants have been widely researched [8]. Xu et al. demonstrated that α-pinene could inhibit the feeding activity of *Dendroctonus valens* and also acted as a precursor of bark beetle’s pheromones [10]. The cytochrome P450 *CYP6DE1* of *Dendroctonus ponderosae* can directly use α-pinene to synthesize *trans*-verbenol, which is a kind of aggregation pheromone [6]. In addition, α-pinene potentially inhibited the nervous system but induced the antioxidant system in *Ephestia kuehniella* [9]. Many studies have also indicated that α-pinenes can be used as insecticides [7,8].

In this study, the expression level of *DaCYP4BQ1* was significantly upregulated in adults after fumigation of (+)-α-pinene from 24 h to 72 h, suggesting that this CYP gene may play a role in reducing the negative impact of terpenoids on beetles. The finding of an upregulation pattern of *DaCYP4BQ1* is in line with that of *CYP4* genes in *T. castaneum, Chironomus tentans*, *Diabrotica virgifera virgifera*, *Solenopsis invicta*, and *A. gossypii* [17,26,40,41,42]. The expression of all genes was indicated to be induced by phytochemicals. RNAi is an effective method to explore the potential physiological function of insect CYP4 subfamily genes in degrading xenobiotics [17,42]. In order to explore the role of *CYP4BQ1*, we first performed a gene silencing program on the identified CYP4 gene. The results showed it could effectively suppress *CYP4BQ1* expression. Furthermore, successfully silencing *CYP4BQ1* significantly inhibited CYP activity and resulted in increased mortality of adults. This suggests that *DaCYP4BQ1* definitely contributes to (+)-α-pinene sensitivity in *D. armandi*, and it provides further proof for the involvement of this P450 gene in detoxication in the beetles. 

At present, seven proposed models describe the theoretical signal pathways induced by phytochemicals of insect P450 [18]. Under normal circumstances, phytochemicals can diffuse into cells, where they can trigger an ROS burst, which may be transformed into increased P450 expression through the CncC pathway [25]. In this study, the transcriptional level of *CncC* induced by exposure to (+)-α-pinene showed that (+)-α-pinene may activate the CncC pathway (Figure 3A). In addition, we also observed two important genes of CncC pathway, *DaKeap1* and *DaMaf* (Figure 3B,C). The increased expression of *Maf* induces the CncC to enter the nucleus, in which it can heterodimerize with Maf [43]. A previous study showed that the CncC and Maf complex regulated the transcriptional levels of P450 genes through the regulation of promoter activities of these genes [19]. These results indicate that this transcription factor may help protect individuals from (+)-α-pinene toxicity.

The studies of xenobiotic detoxification signaling pathways are of great significance for understanding the molecular mechanism of insecticide resistance [27]. The CncC pathway is one of the exogenous substance detoxification pathways, which plays a key role in regulating the detoxification and antioxidant gene expression [26]. In the past few years, six P450-mediated phytochemical resistance mechanisms have been associated with the CncC signaling pathway [24,44,45]. Interestingly, depletion of *CncC* in *D. armandi* adults suppressed *CYP4BQ1* expression and reduced adult tolerance to (+)-α-pinene in our study (Figure 4). Previous RNA-seq analysis indicated a decrease in the transcription level of CYP4 family genes after silencing *CncC* in the pesticide-resistant strain of *T. castaneum* [26]. Similar results have also been reported in other insect species including *L. decemlineata*, *S. litura*, and *T. castaneum* [17,25,26]. Nevertheless, these studies did not test whether silencing CncC inhibited insecticide/phytochemical-mediated P450 gene expression. 

The present results found that silencing of *DaCncC* suppressed this (+)-α-pinene-mediated P450 gene induction, indicating that *DaCYP4BQ1* was regulated by the CncC signaling pathway. We also observed H_2_O_2_ production and enhanced activities of SOD, POD, and CAT in (+)-α-pinene-treated adults, suggesting that these antioxidant enzymes were used by *D. armandi* to minimize oxidative damage by (+)-α-pinene through eliminating excessive ROS. However, it is unclear how H_2_O_2_ accumulation and the oxidative burst affect P450 expression and insecticide/phytochemical tolerance. Furthermore, we performed NAC treatment, which is an ROS scavenger, finding that ROS induced the transcription of P450 genes through activating *CncC* and *Maf*. We also applied curcumin, which is an effective CncC agonist that has been found to activate the expression of 20-hydroxyecdysone synthesis-related P450 genes in *S. litura* and *Bombyx mori* [46,47], to define the roles of CncC in adult tolerance to (+)-α-pinene. The results showed that activation of the CncC signaling pathway via ROS burst participates in regulating the P450 gene responsible for phytochemical tolerance in *D. armandi*. The mechanism of CncC pathway initiation in insects seems to be conservative, as previous studies also found that insecticide λ-cyhalothrin- and eugenol-induced ROS burst induces P450 gene in *S. litura* and *T. castaneum,* respectively [17,25]. These findings will contribute to the development of new therapeutics to pest management. However, whether the other signal pathway also regulates the expression of other detoxification genes in response to (+)-α-pinene will be determined in further research. Moreover, enzymatic conversion of (+)-α-pinene into products with low activity also needs further exploration.

## 4. Materials and Methods

### 4.1. Insects and Reagents Preparation

*D. armandi* insects were collected from infested *P. armandii* at the Huoditang Experimental Forest Station, which is located in the southern slopes of the middle Qinling Mountains, China (33°18′ N, 108°21′ E). Then, they were reared on the artificial food in an artificial climate cabinet at 25 ± 1 °C and 70% relative humidity (RH), in the dark [48]. (+)-α-Pinene (98%) acetone and dimethyl sulfoxide (DMSO) were obtained from Aladdin Industrial. N-Acetylcysteine (NAC) and curcumin were purchased from Shanghai Sangon Biotech Company (Shanghai, China)

### 4.2. Tolerance to (+)-α-Pinene

Fumigation treatment was performed as described in previous research [49]. The males and females of emerged adults were divided into six groups, and then treated with 5 μL of (+)-α-pinene (LC_50_) or acetone (control group) for 2 h in 20 mL glass vial [5]. Each group contained 40 adults with essentially the same size. After the adults regained their vitality, they were transferred to an artificial climate cabinet. To explore the effect of (+)-α-pinene on expression of *DaCYP4BQ1* (GenBank number: KR012835), *DaCYP4BQ3* (KR012824), *DaCYP4G4* (KR012827), and *DaCYP4R4* (KR012825), the surviving adults were collected at 72 h post exposure to LC_50_ (+)-α-pinene. In addition, *DaCYP4BQ1, DaCncC* (ON637116), *DaKeap1*(ON637117), and *DaMaf* (ON637118) expressions were observed at 24, 48, and 72 h post exposure to LC_50_ (+)-α-pinene. At the same time, the acetone-treated surviving adults were collected as controls at the same point. 

### 4.3. RNA Extraction, cDNA Synthesis, and Reverse Transcription Quantitative PCR (RT-qPCR) 

Total RNA was determined as previously described [50]. The relative expression levels of each CYP4 gene were quantified by qRT-PCR. The qRT-PCR was performed as described in our previous study [50]. Specific primers were used to detect the expressions of *DaCYP4s, CncC*, *Keap1*, and *Maf* genes. The *CYP4G55* (accession number: JQ855658.1) and *β-actin* (accession number: KJ507199.1) sequences of *D*. *armandi* were used as the internal control [49,51]. The relative expression levels were analyzed using the 2^−ΔΔCt^ method [52].

### 4.4. Developmental and Tissue-Dependent Expression Profiles of CYP4BQ1, CncC, Maf, and Keap1

*D. armandi* eggs were separated into two substages: early eggs (light color) and late eggs (about to hatch). Larvae were also separated into two substages: early larvae and late larvae. Pupae were separated into two substages: early pupae and late pupae. We separated the adults into three substages: teneral adults, emerged adults, and attacking adults [53]. The antennae, foregut, brain, midgut, fat body, hindgut, pheromone gland, Malpighian tubules, and hemolymph of emerged adults were stored at −80 °C. Three independent biological replicates were prepared for gene expression analysis.

### 4.5. Double-Strand RNA (dsRNA) Synthesis and Injection

The T7 Express RNAi System (Promega, Madison, MI, USA) was used for dsRNA synthesis [54]. RNAi primers (Appendix A) were designed on the basis of the obtained sequences. The survival rates were recorded at 48 h after the injection with dsRNA. After injection, six adults of each treatment were collected at 24, 48, and 72 h for qRT-PCR, with 20 used for CYP activity measurement, which was assessed by an enzyme-linked immunosorbent assay (ELISA) kit (catalog no. H190-1, Nanjing Jiancheng Bioengineering Institute, Nanjing, China). At 48 h after dsRNA injection, the adults were treated with (+)-α-pinene (LC_50_) for 2 h; then, these beetles were reared for 48 h in normal conditions and the mortality was determined. To determine the function of CncC signaling pathway in (+)-α-pinene tolerance of *D. armandi* emerged adults, we also used RNAi to silence CncC, and toxicity analysis was performed as described above.

### 4.6. H_2_O_2_ Content and CAT, SOD and POD Activity Determination

As described above, the surviving adults were collected at 24, 48, and 72 h post exposure to LC_50_ (+)-α-pinene. Meanwhile, the acetone-treated surviving adults were collected as controls at the same timepoint. H_2_O_2_ content and CAT, SOD and POD activities were determined spectrophotometrically with a Biotek microplate reader using commercial kits (CAT: catalog no. A007-1; SOD: catalog no. A001-1; POD: catalog no. A084-1. Jiancheng Bioengineering Institute, Nanjing, China) as previously described [25].

### 4.7. Scavenging of ROS by N-Acetylcysteine (NAC) Treatment

To determine whether ROS scavenger treatment can affect the susceptibility of *D. armandi* adults to (+)-α-pinene, the adults were reared on normal diet as control or 2% NAC-containing diet for 48 h. The adults were further exposed to acetone or (+)-α-pinene using a fumigation treatment method, and mortality was measured 24 h later. Each treatment was tested on 30 adults, with three replicates of each bioassay. To determine the effects of ROS scavenging by NAC ingestion on H_2_O_2_ content and P450 activity in the midgut, the expressions of *CncC, Maf*, and *CYP4BQ1* were measured as described above.

### 4.8. Dietary Curcumin and (+)-α-Pinene Tolerance Analyses

To investigate the role of the CncC signaling pathway in P450-mediated (+)-α-pinene tolerance, we conducted a curcumin-feeding assay as previously described. The adults were reared on normal a diet containing curcumin for 48 h, and then they were exposed to LC_50_ (+)-α-pinene for tolerance analyses (as control). The effects of curcumin application on P450 activity in the midgut and *CncC, Maf*, and *CYP4BQ1* expression were measured as described above.

### 4.9. Statistical Analysis

All data analyses were performed with SPSS Statistics 19.0 (IBM, Chicago, IL, USA). Post hoc Tukey tests were used to check the difference through one-way ANOVA. Student’s *t*-test was performed with two-sample analyses. Prism (GraphPad Software, version 6, USA) were used to plot Graphs.

## 5. Conclusions

Exposure of *D. armandi* adults to the host allelochemical (+)-α-pinene enhanced the activities of antioxidant enzymes and H_2_O_2_ production, which resulted in the elicitation of oxidative burst, with subsequent activation of the CncC signaling pathway that induces *CYP4BQ1* expression. *CYP4BQ1* is responsible for (+)-α-pinene tolerance in *D. armandi*. Collectively, we indicated that *D. armandi* regulates cytochrome P450 *CYP4BQ1* by activating the CncC signaling pathway via elicitation of the ROS burst, thereby responding to (+)-α-pinene. Nevertheless, how (+)-α-pinene elicits this route of detoxication mechanism remains obscure, which need to be further elucidated. The present study explained the regulatory mechanism of P450 gene induction participating in allelochemical tolerance and might provide a novel target for pest control.

## Figures and Tables

**Figure 1 ijms-23-11578-f001:**
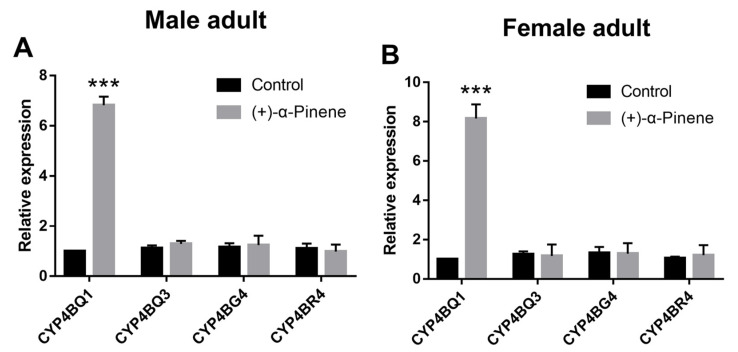
Relative expression levels of four *CYP4B* genes in *D. armandi* emerged male (**A**) and female (**B**) adults after stimulation with LC_50_ (+)-α-pinene at exposure time of 72 h. The asterisk indicates a significant difference between different treatment groups (*** *p* < 0.001, independent Student’s *t*-test). All values are the mean ± SE, *n* = 3.

**Figure 2 ijms-23-11578-f002:**
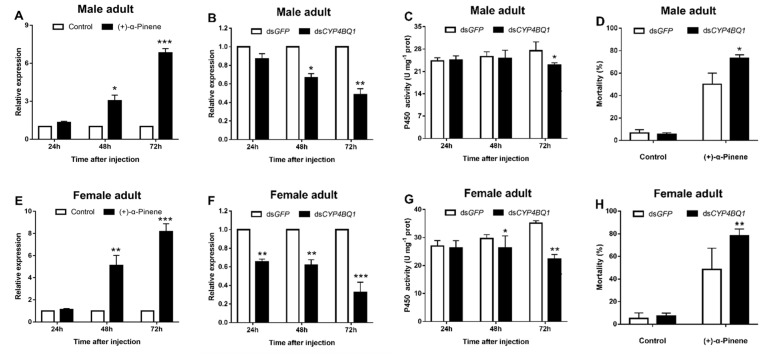
*CYP4BQ1* is involved in (+)-α-pinene detoxification by *D. armandi* adults. The surviving adults were collected at 24, 48, and 72 h post exposure to LC_50_ (+)-α-pinene. (**A**,**E**) Induction of *CYP4BQ1* expression by (+)-α-pinene treatment. (**B**,**F**) Repression of *CYP4BQ1* expression in *D. armandi* adults injected with dsRNA. (**C**,**G**) P450 activities were determined after RNAi. (**D**,**H**) The mortality of adults exposed to LC_50_ (+)-α-pinene was assessed at 48 h after dsRNA injection. The asterisk indicates a significant difference between treatment groups (* *p* < 0.05, ** *p <* 0.01, *** *p* < 0.001, independent Student’s *t*-test). All values are the mean ± SE, *n* = 3.

**Figure 3 ijms-23-11578-f003:**
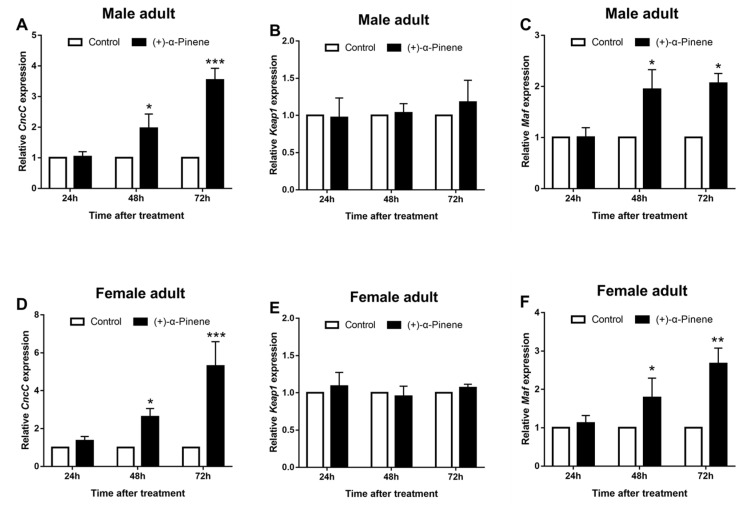
Relative expression levels of *CncC* (**A**,**D**), *Keap1* (**B**,**E**), and *Maf* (**C**,**F**) in *D. armandi* emerged adults (sex separated) after stimulation with LC_50_ (+)-α-pinene at exposure times of 24, 48, and 72 h. The asterisk indicates a significant difference between treatment groups (* *p* < 0.05, ** *p <* 0.01, *** *p* < 0.001, independent Student’s *t*-test). All values are the mean ± SE, *n* = 3.

**Figure 4 ijms-23-11578-f004:**
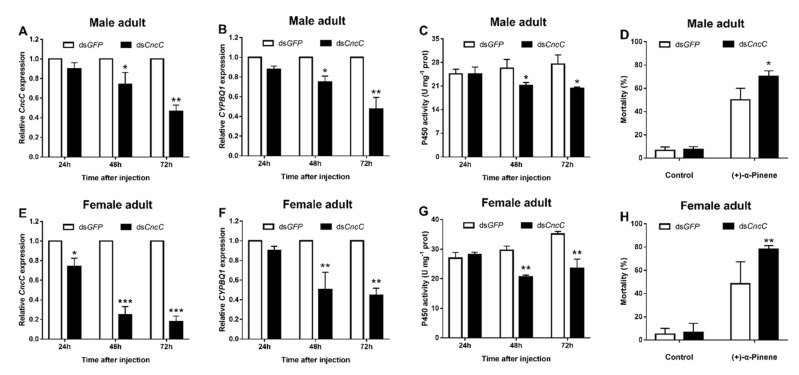
Silencing *CncC* suppresses *CYP4BQ1* expression and reduces adults’ tolerance to (+)-α-pinene. Adults of *D. armandi* were treated with dsRNA for 24, 48, and 72 h after injection. The expression levels of *CncC* (**A**,**E**) and *CYP4BQ1* (**B**,**F**) were detected by RT-qPCR. The P450 activities of midgut (**C**,**G**) were determined after RNAi. The mortality (**D**,**H**) of adults exposed to LC_50_ (+)-α-pinene was assessed at 48 h after dsRNA injection. The asterisk indicates a significant difference between treatment groups (* *p* < 0.05, ** *p <* 0.01, *** *p* < 0.001, independent Student’s *t*-test). All values are the mean ± SE, *n* = 3.

**Figure 5 ijms-23-11578-f005:**
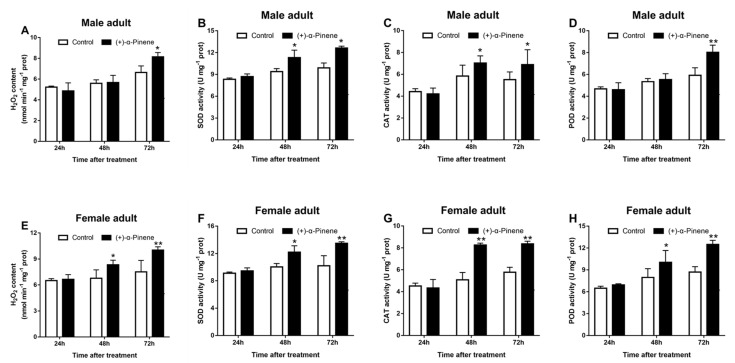
Effects of (+)-α-pinene exposure on H_2_O_2_ generation and the activities of antioxidant enzyme in *D. armandi.* The surviving adults were collected at 24, 48, and 72 h post exposure to LC_50_ (+)-α-pinene. The H_2_O_2_ content (**A**,**E**) and the activities of SOD (**B**,**F**), CAT (**C**,**G**), and POD (**D**,**H**) in the midgut of adults were determined spectrophotometrically. The asterisk indicates a significant difference between different treatment groups (* *p* < 0.05, ** *p <* 0.01, independent Student’s *t*-test). All values are the mean ± SE, *n* = 3.

**Figure 6 ijms-23-11578-f006:**
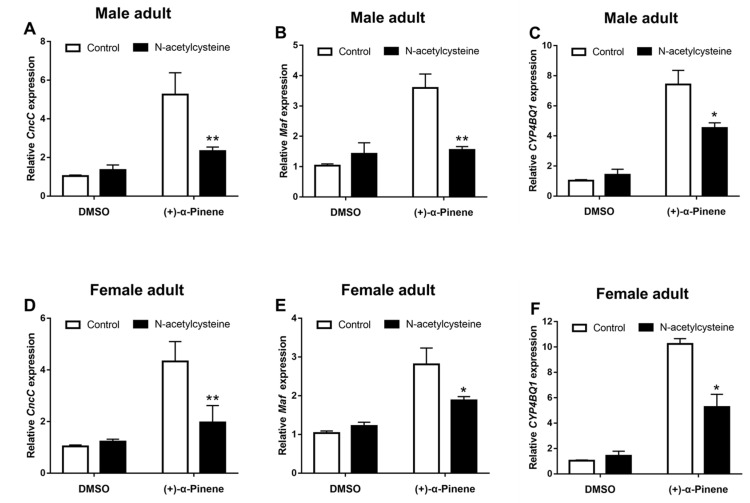
Effects of ROS scavenging on the expression of (+)-α-pinene-induced *CncC* (**A**,**D**), *Maf* (**B**,**E**), and *CYP4BQ1* (**C**,**F**) genes. The adults of *D. armandi* reared on normal (control) or 2% NAC diets were treated with DMSO or (+)-α-pinene for 24 h. The asterisk indicates a significant difference between treatment groups (* *p* < 0.05, ** *p <* 0.01, independent Student’s *t*-test). All values are the mean ± SE, *n* = 3.

**Figure 7 ijms-23-11578-f007:**
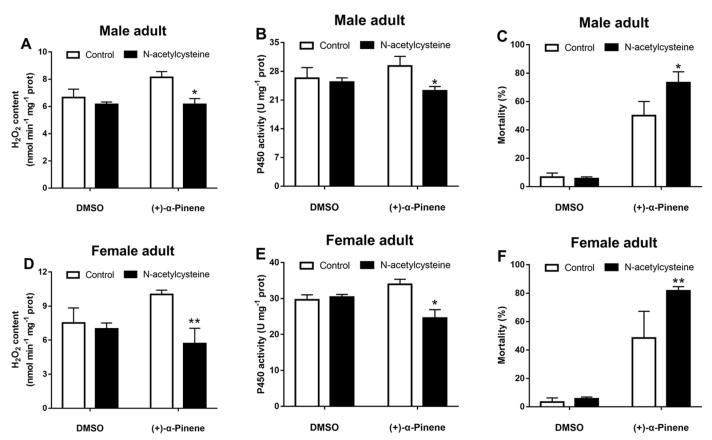
Effects of ROS scavenging on the adults’ tolerance to (+)-α-pinene in *D. armandi.* The adults of *D. armandi* reared on normal (control) or 2% NAC diets were treated with DMSO or (+)-α-pinene for 24 h. The H_2_O_2_ content (**A**,**D**) and the activities of P450 (**B**,**E**) in the midgut of adults were determined spectrophotometrically. (**C**,**F**) Adults were further treated with (+)-α-pinene, and mortalities were assessed after 24 h. The asterisk indicates a significant difference between different treatment groups (* *p* < 0.05, ** *p <* 0.01, independent Student’s *t*-test). All values are the mean ± SE, *n* = 3.

**Figure 8 ijms-23-11578-f008:**
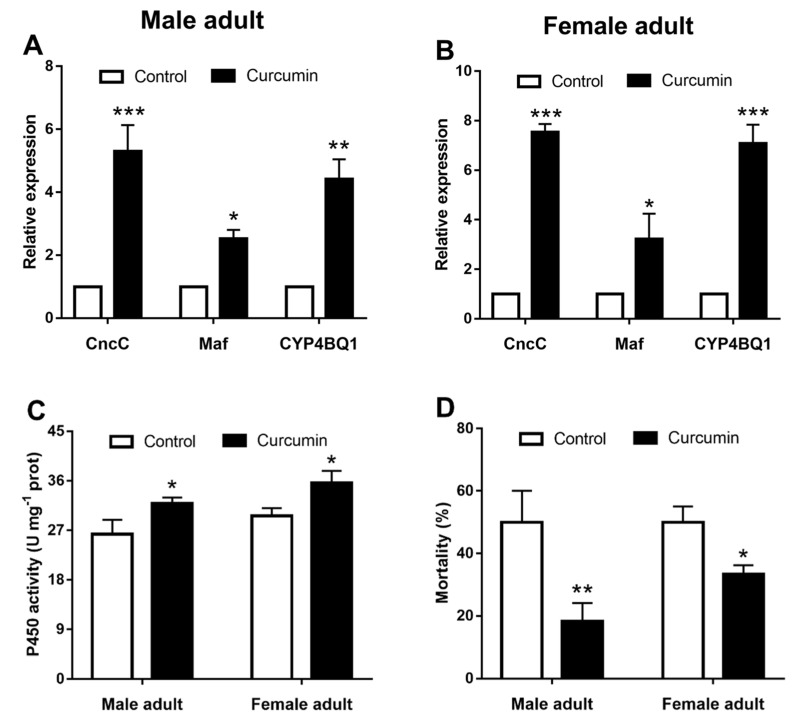
Activation of CncC pathway by the curcumin induces the expression of *CYP4BQ1* and enhances adults’ tolerance to (+)-α-pinene. Adults were treated with the CncC agonist curcumin for 48 h. The relative expression levels of *CncC, Maf*, and *CYP4BQ1* (**A**,**B**) were detected by RT-qPCR. The activities of P450 (**C**) in the midgut of adults were determined spectrophotometrically. (**D**) Curcumin-treated adults were then exposed to (+)-α-pinene, and mortality was assessed after 24 h. The asterisk indicates a significant difference between treatment groups (* *p* < 0.05, ** *p <* 0.01, *** *p* < 0.001, independent Student’s *t*-test). All values are the mean ± SE, *n* = 3.

## Data Availability

The data are available from the corresponding author on reasonable request.

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
