# Peer review of "Activation of the ROS/CncC Signaling Pathway Regulates Cytochrome P450 CYP4BQ1 Responsible for (+)-α-Pinene Tolerance in Dendroctonus armandi"

_ijms, 2022, doi:10.3390/ijms231911578_

Round 1

Reviewer 1 Report

The manuscript authored by Liu et al. to IJMS focused on the mechanism of how the bark beetles rely on one P450 to resist the ?-pinene from host tree. In this study, the authors found that host tree defensive component ?-pinene-induced ROS activated transcription factors CncC and Maf, and finally upregulated P450 CYP4BQ1 to detoxify. Overall, the study provides more information on the interactions between insects and host plants and more details on the insect strategy to overcome toxic compounds from host tree. The study was well designed and conducted, the interesting results may provide a new target for pest control. I only have several comments that may improve the manuscript:

1.       The ?-pinene induced the production of ROS, and then triggered the following reactions, while scavenging of ROS by NAC downregulated the reactions. If you would like to use H2O2 to mimic the ?-pinene induced ROS and test the following gene changes, the NAC lose function assay and H2O2 gain function assay will make the results more reliable. 

2.       Some information in ref 17, 41 and 43 is missing, please double check all the ref.

3.       A model to summarize all the results will make the manuscript more attractive.

4.       In figure 2, 3 and 4, some control groups in different time points are same, but others are not, could you explain the difference?

5.       Line 102, please delete “IN”.

6.       Please detail the role of P450 on ?-pinene catabolism in line 53.

7.       In line 124, 159, 170, 188, 202, 217, 224 and 239, please replace “Student’s test” with “Student’s t-test”.

8.       Line 408, please list the commercial kits catalog number.

9.       Line 10, font size is not uniform.

Author Response

Dear Reviewer:

Thanks for your comments, I have provided a point-by-point response to you. Please see the attachment.

Reviewer 2 Report

Bin Liu et al. ‘Activation of the ROS/CncC signaling pathway regulates cytochrome P450 CYP4BQ1 responsible for (+)-α-pinene tolerance in Dendroctonus armandi

I did not have access to the Supplementary Information, so the comments below assume that Figs. S1 and S2 show what the authors claim they do.

The Chinese white pine beetle is a significant pest of Pinus armandii and new strategies are being sought to control this pest. When attacked the trees release defensive volatiles, the major component of which is α-pinene. α-Pinene is toxic to the beetles, unless the beetles are able to detoxify it. The authors have begun to elucidate how D. armandi responds to exposure to α-pinene and have found that expression of the cytochrome P450 CYP4BQ1 gene is significantly enhanced after exposure of the insects to α-pinene and they present evidence that regulation of the gene’s activity involves the ROS/CncC signalling pathway.

The approach used by the authors is not fully novel, as it has previously been used with other insect pest species, but it is appropriate for this study. The authors appear to have focussed their studies only on CYP4B genes and only the ROS/CncC signalling pathway. If they have examined other possibilities, they should state whether those contribute to the detoxification of α-pinene and, conversely, if they have not investigated other possibilities, this should also be clearly stated. The authors present convincing evidence that CYP4BQ1 is involved in the detoxification of α-pinene, but the enzymic conversion of α-pinene to a less active product is not described.

The general structure and presentation of the manuscript is good, but the English does need quite a lot of grammatical and stylistic improvement. The Discussion is too long as much of it repeats what is stated in the Results section. The authors claim in the Abstract and Conclusions that this research could lead to new control methods for the pest, but this aspect is not considered, even in outline, in the Discussion. It seems that references [15} and [16] are cited the wrong way round.

Author Response

(The authors gave the same response as above.)

Reviewer 3 Report

CYP enzymes consisting of numerous subtypes make  the major metabolic, both catlytic and anabolic, regulations in organisms including synthesis and detoxication in xenobiotic  conditions. This research focussed on the DaCYP4BQ1 in Dendroctonus armandi to dfend against pinine. It also analyzed the up-stream regulation of ths enzyme induction due to reactive oxigen. Their logic is straight forward and methodology is sound and overall writing is clear. I do not hesitate to support eventual publication, though English need slight polishing. The mechanism how this CYP species is mobilized upon pinene exposure was elucidated relatively clearly but further upstream event, i.e., how pinene elicits this route of detoxication mechanism remains obscure. Mention is needed about this aspect. 

Author Response

(The authors gave the same response as above.)
